# Road Speed Prediction Scheme by Analyzing Road Environment Data

**DOI:** 10.3390/s22072606

**Published:** 2022-03-29

**Authors:** Jongtae Lim, Songhee Park, Dojin Choi, Kyoungsoo Bok, Jaesoo Yoo

**Affiliations:** 1Department of Information & Communication Engineering, Chungbuk National University, Cheongju 28644, Korea; jtlim@chungbuk.ac.kr (J.L.); shpark1586@chungbuk.ac.kr (S.P.); 2Department of Computer Engineering, Changwon National University, Changwon 51140, Korea; dojinchoi@changwon.ac.kr; 3Department of Artificial Intelligence Convergence, Wonkwang University, Iksan 54538, Korea; ksbok@wku.ac.kr

**Keywords:** road speed prediction, traffic congestion, traffic incident analysis, traffic prediction, traffic data analysis

## Abstract

Road speed is an important indicator of traffic congestion. Therefore, the occurrence of traffic congestion can be reduced by predicting road speed because predicted road speed can be provided to users to distribute traffic. Traffic congestion prediction techniques can provide alternative routes to users in advance to help them avoid traffic jams. In this paper, we propose a machine-learning-based road speed prediction scheme using road environment data analysis. The proposed scheme uses not only the speed data of the target road, but also the speed data of neighboring roads that can affect the speed of the target road. Furthermore, the proposed scheme can accurately predict both the average road speed and rapidly changing road speeds. The proposed scheme uses historical average speed data from the target road organized by the day of the week and hour to reflect the average traffic flow on the road. Additionally, the proposed scheme analyzes speed changes in sections where the road speed changes rapidly to reflect traffic flows. Road speeds may change rapidly as a result of unexpected events such as accidents, disasters, and construction work. The proposed scheme predicts final road speeds by applying historical road speeds and events as weights for road speed prediction. It also considers weather conditions. The proposed scheme uses long short-term memory (LSTM), which is suitable for sequential data learning, as a machine learning algorithm for speed prediction. The proposed scheme can predict road speeds in 30 min by using weather data and speed data from the target and neighboring roads as input data. We demonstrate the capabilities of the proposed scheme through various performance evaluations.

## 1. Introduction

Various studies have recently been conducted to solve problems caused by traffic congestion [1,2,3,4,5,6,7,8,9,10,11,12,13,14,15,16,17]. These studies have aimed to reduce the occurrence rate of traffic congestion by predicting traffic congestion in advance and avoiding various problems caused by traffic congestion by providing alternatives to drivers approaching traffic jams. Road speed is one of the most important indicators of traffic conditions. Various factors affect road speed, including the speed limit of a road, traffic volume that the road can accommodate, traffic flow over time, and the effects of connected roads, accidents, weather, and special days such as national holidays. These factors affecting road speed are defined as road environment data. Because road environment data affect traffic congestion, it is necessary to analyze the impact of each factor and combined factors on traffic congestion.

The degree of traffic congestion is determined by various factors such as road speed, traffic volume, number of low-speed vehicles, and road congestion. The National Intelligent Transport System (ITS) Center, which manages all traffic information in South Korea, uses road speed as an indicator for determining traffic conditions. Based on road speed predictions, users can predict road congestion events to avoid congestion and find alternatives in advance, allowing them to avoid various problems caused by traffic congestion. Furthermore, predicted road speeds can be used by navigation services to predict driving time, which is the total travel time from a departure location to a destination.

In the past, road speed prediction schemes have mainly considered speed and traffic volume data [5,6]. Existing schemes use speed and traffic volume changes over time to predict the normal flow on a road segment. The authors of [6] used a Bayesian network [18] to predict traffic speed. They considered the traffic conditions of upstream/downstream roads around the target road to improve prediction accuracy. The authors of [15] used a long short-term memory (LSTM) [19] model to predict short-term road speeds. They used historical speed data from neighboring roads connected to the target road as input data for LSTM. Because the speed data on neighboring roads affect the speeds on connected target roads, they can be used to predict the speed of the target road in the near future. The authors of [7] used the number of low-speed vehicles to determine the degree of road congestion because low-speed vehicles are known to be a key factor in road congestion. The authors of [8] used a convolutional neural network (CNN) [20] model as a tool for predicting road congestion. To predict road speed, they predicted the level of traffic congestion by comprehensively considering the effects of connected roads, traffic accidents, traffic control, and weather.

However, these existing schemes have several limitations. For example, schemes using speed and traffic volume changes and schemes that predict traffic congestion using the number of low-speed vehicles do not consider road environment data [5,7]. The authors of [6,15] considered the effects of the connected roads, but the prediction error increased as the prediction time moved further away from the current time. Additionally, the prediction error gradually increased if an unexpected event occurred.

Furthermore, they did not consider road environment data such as weather and traffic accidents. Prediction schemes using CNN models predict traffic congestion by considering historical traffic congestion data and road environment data [8]. The authors of [8] normalized road environment data to values between zero and one and used the normalized data as inputs for a fully connected neural network to consider road environment data. However, this approach is limited in that it does not provide a scheme for calculating the characteristics of each type of road environment data quantitatively.

Furthermore, most related studies have not considered the impacts of weather and accidents on road speed [4,5,6,7,9,12,13,15]. For example, on very rainy days, the speed of most vehicles decreases significantly. Additionally, if a traffic accident occurs on a road, it takes a certain amount of time to recover the typical road conditions. Therefore, the impacts of weather and traffic accidents on road speed should be considered. In this paper, we propose a machine-learning-based road speed prediction scheme using road environment data analysis. The proposed scheme uses the road speed prediction scheme proposed in [15] to reflect the impact of connected roads. Additionally, the proposed scheme predicts road speed by considering the impact of weather on road speed. If an unexpected event such as a traffic accident or road construction work that breaks the regular flow on the target road occurs, road speed prediction error will increase significantly.

Therefore, the proposed scheme reduces prediction errors by reflecting event weights on road segments where the traffic flow changes rapidly. When predicting the road speed using LSTM, there is a characteristic that the data at the time of prediction have the greatest impact on road speed prediction. Therefore, although the LSTM scheme generally predicts road conditions accurately in the presence of unexpected events, there are cases in which it may fail to predict the regular flow of a road. To address this problem, the proposed scheme analyzes historical average road speed data and uses these data for road speed prediction to reflect the regular flow of a road more accurately.

The remainder of this paper is organized as follows. Section 2 discusses related studies and Section 3 describes the proposed machine-learning-based road speed prediction scheme that considers road environment data. Section 4 shows performance evaluations to verify the capabilities of the proposed scheme. Finally, Section 5 summarizes our conclusions.

## 2. Related Work

### 2.1. Road Congestion Prediction Schemes Using CNNs

Various studies have been conducted to address the problems caused by traffic congestion [3,4,5,6,7,8,9,10,11,12,13,14,15]. Various factors affect road congestion. Existing road congestion prediction schemes consider historical road congestion, the effects of connected roads, weather, accidents, special days, and congestion caused by police to predict the congestion levels of roads [8]. The authors of [8] used a CNN [20] model to consider the spatial characteristics of roads affected by connected roads and the temporal characteristics of congestion levels that change over time.

Figure 1 shows the overall processing structure of the road congestion prediction model proposed in [8]. The snapshot in the upper-left corner of Figure 1 shows the input data used in [8]. The authors constructed a two-dimensional array containing the congestion levels of the road to be predicted and those of the adjacent roads to reflect the impact of adjacent roads and traffic flow over time. They considered the two-dimensional array as a snapshot, which was used as the input for a CNN. Additionally, they normalized data related to weather, holidays, road construction, and special days to reflect the factors affecting congestion. These normalized data and predicted road conditions were used as inputs for a fully connected neural network to predict the final road congestion level.

### 2.2. Road Speed Prediction Scheme Using a Bayesian Network

Road speed is an important indicator for determining the degree of traffic congestion. The authors of [6] used a Bayesian network to predict road speed. They considered the speeds of the upstream and downstream roads around the target road as factors affecting the speed of the target road. They designed a Bayesian network with conditional events in which the road speed prediction for a particular node was determined by the latest information on nearby links. Additionally, because it is difficult to obtain sufficient data on unanticipated events for model training, they predicted final road speeds by reflecting real-time road speeds to resolve performance degradation issues.

Figure 2 shows a road network and Bayesian network designed to predict road speeds. A line segment represents a road and an arrow represents the flow of a road. Assuming that the Bayesian network is constructed to predict the speed on line segment BC, the resulting node in the Bayesian network in Figure 2 is BC(t). There are three types of links in the network. The latest road speeds on CD, CG, and CH represent upstream links. The latest road speeds on AB, EB, and FB represent downstream links. BC is the target link. To reflect the effects of connected upstream/downstream links, they designed a Bayesian network model, as shown in Figure 2, and predicted road speeds. Additionally, when the prediction error was large in the prediction results obtained by the real-time Bayesian network as a result of an unexpected event, they determined that the current traffic event exhibited a different pattern compared to the typical flow. In such cases, they predicted the final road speed by reflecting the real-time traffic information of an unexpected event with a high weight, rather than simply using the result predicted by the Bayesian network.

### 2.3. Road Speed Prediction Scheme Using LSTM

In ref. [15], authors used an LSTM [19] model to predict road speeds five minutes into the future. Their goal was to improve the accuracy of the predicted speed by analyzing the factors that affect road speed. They performed prediction by independently reflecting the historical speed data from the target road, effects of changes in the traffic volume on road speed, and effects of the speeds of connected roads on the speed of the target road. As a result of calculating and comparing the root-mean-squared errors (RMSEs) between the predicted speeds and actual speeds, they determined that speed prediction considering the effects of the speeds of connected roads yields the highest accuracy.

Figure 3 shows a graph comparing the actual speeds and predicted speeds considering the effects of the speeds of connected roads from [15]. The red and blue lines indicate the actual and predicted speeds, respectively. This scheme yields very high prediction accuracy because unlike the scheme proposed in [6], it trains a prediction model based on historical data and uses the data up to the prediction time point as input data for machine learning. The experimental results from [15] demonstrate that speeds in the near future can be predicted with high accuracy, even without considering road environment data such as weather and road conditions that affect the road speed or traffic accidents that trigger the sharpest changes in road speed. This is because this scheme considers the data up to the time point just before prediction as inputs. When prediction is performed using data up to the previous time point, the accuracy of prediction improves because road speeds typically change gradually.

### 2.4. Problems Faced by Existing Schemes

Ref. [8] improved the accuracy of congestion prediction by using historical road congestion data and data such as weather, road construction, and special day data, which are factors affecting road speed. They normalized each type of road environment data to values between zero and one. Since the road environment data exhibit different characteristics, the impact of each type of data on road congestion must be analyzed. However, they did not provide a scheme for calculating the influence of each data type quantitatively.

Ref. [6] used historical speed data and real-time traffic information from upstream and downstream roads around the target road based on a Bayesian network to predict road speeds after 5 to 60 min. However, in the scheme proposed in [6], the error increased as the prediction time moved further away from the current time. Although this scheme reflects real-time traffic information to prepare for the impact of unanticipated events on road speed, the error increases as the predicted time moves further into the future because the effectiveness of real-time traffic information decreases.

Ref. [21] used a recurrent neural network (RNN) to predict road congestion. However, RNNs face the vanishing gradient problem, where the impact of the initial data disappears as the time gap increases.

Ref. [19] predicted road speeds in the near future by using the historical speed data from roads connected to the target road as inputs for an LSTM model to consider the effects of connected roads. However, when predicting the road speed in the relatively distant future, there is a problem where the prediction error increases if an unanticipated event occurs.

Various factors affect road speed and various road environment data should be considered to increase the accuracy of road speed prediction. Because roads are all connected and connected roads have mutual impacts, the effects of connected roads should also be considered. Particularly, factors such as weather and traffic accidents have a large impact on speed. Because road environment data have different characteristics and influences, the characteristics of each type of road environment data should be analyzed and used for road speed prediction to increase the accuracy of road speed prediction.

## 3. Proposed Road Speed Prediction Scheme

### 3.1. Overall Processing Approach

As the cost of traffic congestion has increased, various studies have been conducted on the prediction of traffic congestion and road speed to address this problem [22,23]. The National ITS Center uses road speed as an indicator to identify various traffic events. Drivers can avoid congestion by anticipating congestion events based on road speed predictions. Additionally, they can avoid various problems caused by traffic congestion by receiving alternative routes in advance. Existing road speed prediction schemes use the latest road speed data to supplement the problem of a significant decline in prediction accuracy when an unexpected event occurs. However, when the prediction time is far in the future, there is a limitation when an unexpected event occurs between the current time point and prediction time point, even if the latest road speed data are reflected. Therefore, road speed prediction schemes should consider road environment data that can reflect unexpected events. Existing road congestion prediction schemes that use CNN models consider various road environment data, but they do not provide a scheme for calculating the impacts of road environment data with different characteristics quantitatively.

In this paper, we propose a road speed prediction scheme that considers road environment data to resolve the problems faced by existing schemes. The proposed scheme uses historical speed data and historical average speed data for the target road. The proposed scheme also considers weather data and historical speed data from connected roads as road environment data. Furthermore, it analyzes speed changes to consider unexpected events such as accidents and construction work that cause rapid changes in road speed. Therefore, the proposed scheme can overcome the problems of existing schemes, where the accuracy of the predicted road speed decreases when unexpected events occur such as bad weather, accidents, or construction.

Figure 4 shows the overall structure of the proposed road speed prediction scheme. The processes are divided into offline processing, which is the training stage, and online processing, which is the prediction stage. Online processing generates a dataset through the normalization of data collected in real time. It then uses the generated prediction dataset as input data for a trained LSTM model. The prediction dataset consists of speed data from the target road and neighboring roads up to the current time point, as well as forecasted precipitation data for the next 30 min.

The proposed scheme uses the prediction dataset to predict a primary speed that reflects the effects of weather and neighboring roads. However, because the primary predicted speed result does not consider the effects of unexpected events, the typical flow of the target road is sometimes not predicted accurately. This is because the latest data have a relatively large impact on prediction. To overcome the problem of poor accuracy caused by unexpected events, the proposed scheme also considers event weights that can reflect rapidly changing flows on the road caused by unexpected events in the primary predicted speed. Finally, the proposed scheme corrects the error of the primary predicted speed based on historical average speed data to predict a final speed. Offline processing trains a prediction model by inputting a training dataset into an LSTM model. The training dataset consists of precipitation data and historical average speed data for the target road and neighboring roads. The proposed scheme uses a learning model that applies optimal weights by comparing actual and predicted speeds through the learning process.

### 3.2. Normalization

Prediction schemes using neural network models are highly accurate. However, the form of data is very important for learning schemes using neural network models. Therefore, the proposed scheme normalizes the precipitation data from the pertinent region and speed data of the target and neighboring roads collected in 5 min intervals as inputs for the LSTM model. The goal of this procedure is to make the original data suitable for neural network model training. Data sometimes contain missing values. Missing values can significantly reduce the performance of prediction models. Missing values can be removed or corrected using various filling schemes. In the initially collected speed data, missing values were given values of 0.0. However, based on the nature of the road speed data, road speed over time is a factor that cannot be ignored. Therefore, the proposed scheme uses the fillna() function provided by the Python Pandas library [24] to fill in missing values with the average values of the neighboring values. If consecutive values are missing, the proposed scheme fills the missing values using the average speed values of the pertinent road from the same day and time in the past.

The input data for the neural network model must be normalized independently according to the characteristics of each dataset [25]. Therefore, the proposed scheme normalizes the speed and weather data according to Equations (1) and (2), respectively. For easy training of the neural network model, each data point is typically normalized to a value between zero and one. This means that the characteristics of all the data should be relatively uniform to fit within similar ranges. In Equation (1), $S_{R}$ denotes the overall speed data collected from road *R*. $S_{R}^{t}$ denotes the speed at time t collected from road *R*. To reflect the general characteristics of each road, the proposed scheme divides the speed data collected from the target road by the largest value among the speed data from each road, resulting in normalization to a derived value between zero and one. For example, assuming that $\underset{\;}{\max}\left(S_{R}\right)$ is 110 and $S_{R}^{t}$ is 89, the normalized value of $S_{R}^{t}$ is 0.81. Equation (2) is the normalization equation for the weather data. $Rainfall_{T}^{t}$ denotes the rainfall in the region to which the target road belongs at hour t. Similar to speed data normalization, the proposed scheme normalizes each rainfall value to a value between zero and one by dividing it by the maximum value among the collected weather data.
(1)$$nomalized\;S_{R}^{t}=S_{R}^{t}/\underset{\;}{\max}\left(S_{R}\right)$$
(2)$$nomalized\;Rainfall_{T}^{t}\;\equiv \;Rainfall_{T}^{t}/\underset{\;}{\max}\left(Rainfall_{T}\right)$$

### 3.3. Generation of a Dataset

Road speed prediction requires a training dataset to train a prediction model, as well as a dataset for actual prediction. Table 1 shows an example of a prediction dataset used for data preprocessing that is computed by Equations (1) and (2). $S_{T}^{i}$ and $S_{N_{n}}^{i}$ denote the data from the target road and a neighboring road, respectively, at time point *i*. $Prediction\;Rainfall_{T}^{i+6}$ (hereafter $PRF_{T}^{i+6}$) reshows the expected rainfall at time point *i* + 6. All data were recorded at 5 min intervals. When performing prediction, because the actual road speed and rainfall after 30 min cannot be known, a prediction dataset must be generated.

Table 2 shows an example training dataset. A training dataset is used to train a prediction model by repeating the process of comparing the predicted and actual road speeds after 30 min. Because the proposed scheme uses past data for training, the actual speed and rainfall at time point *i* + 6 can be obtained. Therefore, the proposed scheme provides the actual speed data on the target road at time points *i* and *i* + 6, and the actual rainfall $Rainfall_{T}^{i+6}$ in the same row of the data matrix, as shown in Table 2. The process of training a prediction model is described in detail in the following section.

### 3.4. Training of a Prediction Model

The proposed scheme trains an LSTM model offline to predict road speeds based on the effects of neighboring roads and weather online. Figure 5 shows the LSTM model training process in the proposed scheme. The LSTM model consists of LSTM, dropout, and dense layers. In the proposed scheme, a training dataset in the format shown in Table 2 is used as the input for the LSTM layer. $S_{T}^{t}$ and $S_{N_{n}}^{t}$ denote the data of the current target road and neighboring roads, respectively. $Rainfall{\left(R\right)}_{T}^{t+6}$ denotes the rainfall data at time point t + 6 collected from the region in which the target road is located. The proposed scheme uses these data to reflect the impacts of data from the target and neighboring roads 30 min in the past and rainfall at the predicted time point on the speed of the target road in the prediction model.

The actual road speed ($S_{T}^{i+6}$) after 30 min from the training dataset is used to calculate loss. To prevent overfitting, we incorporated a dropout layer with a dropout rate of 0.3. The model is set to output one result through a dense layer that connects all input and output neurons. $PS_{T}^{t+6}$ denotes the predicted speed of the target road. The proposed scheme uses a loss function of MSE to calculate the error between $PS_{T}^{t+6}$ and $S_{T}^{t+6}$, as shown in Equation (3). The proposed scheme uses Adam as an optimization function to perform training quickly and stably. The proposed scheme performs a back-propagation process to determine the optimal weights for minimizing the MSE. The proposed scheme repeats this process to train a prediction model with optimal weights.
(3)$$MSE=\frac{1}{n}\overset{n}{\underset{i=1}{\sum }}{\left(S_{T}^{t+6}-PS_{T}^{t+6}\right)}^{2}$$

### 3.5. Primary Speed Prediction

For offline processing, the proposed scheme trains a model with optimal weights based on the rainfall and speed of neighboring roads, which are road environment data affecting the speed of the target road. In the primary speed prediction stage, the LSTM model discussed in Section 3.4 is used to predict the road speed after 30 min. The speed predicted at this stage is defined as the primary predicted speed (or primary speed). For primary speed prediction, the predicted scheme uses a prediction dataset consisting of speed data from the target and neighboring roads, as well as the expected rainfall data after 30 min. The proposed scheme predicts the road speed after 30 min by inputting the prediction dataset and optimal weights into the trained LSTM model.

Figure 6 shows a graph of actual road speeds and primary predicted speeds over time. The road speed was not affected by rainfall on 9 September 2019 because it did not rain on that day. The scheme proposed in [15] yields high accuracy when predicting the road speed in the near future but exhibits large errors when predicting the road speed in the relatively distant future. As shown in Figure 6, the scheme proposed in [15] yields a similar speed pattern to that observed 30 min in the past. The scheme proposed in [15] exhibits large differences between the actual and predicted speeds because it uses a gentle curve to reflect the rapidly changing flow of road speed caused by unexpected events and during rush hour. In the primary speed prediction stage, only the effects of neighboring roads and rainfall are considered. Factors that rapidly change road conditions should be considered to improve accuracy. Therefore, the proposed scheme derives final predicted speeds by analyzing historical average road speeds and road speed changes to correct the primary predicted speed.

### 3.6. Correction of Predicted Speed

The primary predicted speed obtained by the model discussed in Section 3.5 exhibits a trend of following the actual speed with a delay because the speed data from the previous time point have the greatest influence. As a result, the error is large in sections where rapid changes occur in the road speed, such as rush hour, because the speed is predicted using a gentle curve. Therefore, in this paper, we propose two schemes for correcting the predicted speed to improve prediction accuracy. The first scheme uses historical average speed data to correct the predicted speed. The proposed scheme considers the average flow of the target road based on historical average speeds for speed prediction. The second scheme corrects the predicted speed by applying event weights based on the analysis of speed changes caused by events such as a traffic accidents or road construction. The proposed scheme considers uncommon road flows based on the application of event weights to perform speed prediction.

#### 3.6.1. Application of Historical Average Speeds

Road speeds exhibit similar trends according to the day of the week and time of day. Figure 7 shows the results of calculating the average speed of the target road for each day of the week and time of day for approximately three months from 20 May to 25 August 2019. The green and red lines indicate the flows of average road speeds on weekdays and weekends, respectively. On weekdays, congestion occurs when the speed decreases sharply between 6:00 and 9:00 a.m. as a result of the effects of road congestion during the morning rush hour. In general, low speeds can also be observed in the evening rush hour. On weekends, there are no notable congestion sections because there are no effects of morning and evening rush hours, resulting in showing smoother flows on the road compared to the weekdays. Rapid changes in flow are difficult to predict using the scheme described in Section 3.4. Therefore, when predicting weekday speeds, the effects of morning rush hour congestion are reflected when correcting the primary predicted speed obtained using the scheme discussed in Section 3.4. First, the proposed scheme defines a section in which the speed decreases based on historical average speed changes. Next, the proposed scheme compares the historical average speed and primary predicted speed in this section and incorporates the historical average speed into the prediction according to the road conditions.

A section in which the speed decreases on average in the past is defined based on the statistics of historical average speed changes over time. Historical speed changes are calculated using Equation (4). The term $\Delta AS_{day}^{t}$ denotes the historical average speed change on a specific day of the week. The term $AS_{day}^{t-3}$ reshows the historical average speed 15 min before time t on a specific day of the week. If $\Delta AS_{day}^{t}$ is greater than zero, it indicates that the speed decreased over the 15 min interval. Conversely, if it is smaller than zero, it indicates that the speed increased during the 15 min interval. Figure 8 shows historical speed changes during the rush hours for each day of the week. On average, the average speed change on every day of the week increases in the positive direction starting at 5:40 a.m. Then, starting at 6:30 a.m., the speed change decreases in the negative direction. In this case, the proposed scheme defines 5:40 to 6:25 a.m. on weekdays as a section in which the historical average speed is incorporated.
(4)$$\Delta AS_{day}^{t}=AS_{day}^{t-3}-AS_{day}^{t}$$

In the above case, when the average speed decrease section is defined from 5:40 to 6:25 a.m., the proposed scheme determines whether the historical average speed should be applied to the defined section using Equation (5). $PS_{day}^{t}$ denotes the primary predicted speed at a particular time on a particular day of the week. $AS_{day}^{t}$ denotes the historical average speed at a particular time on a particular day of the week. $APS_{day}^{t}$ denotes the predicted speed to which the historical average speed may be applied. If the predicted $PS_{day}^{t}$ speed is greater than $AS_{day}^{t}$ in the time band during which the speed decreases on average in the past data, then the proposed scheme replaces the primary predicted speed with $AS_{day}^{t}$. Otherwise, the proposed scheme determines that an uncommon flow is present on the target day and uses the primary predicted speed without modification.
(5)$$AS_{day}^{t}<PS_{day}^{t}=\{\begin{array}{c} True\;:\;APS_{day}^{t}=AS_{day}^{t}\\ False\;:\;APS_{day}^{t}=PS_{day}^{t} \end{array}$$

#### 3.6.2. Application of Event Weights

The proposed scheme improves accuracy by applying historical average speeds to correct primary predicted speeds. However, in the event of an accident or construction, the road speed decreases sharply, representing an uncommon flow. To reflect such a flow, the proposed scheme uses accident and construction work data recorded in real time. To use accident data, the proposed scheme must read accident information recorded in real time and reflect the accident location and accident type in its road speed predictions. However, the time at which an accident occurs and the time at which it is recorded are different and many accident data are inaccurate. Therefore, this paper proposes a road speed prediction scheme that can reflect uncommon flows on a road by applying event weights to sections in which the road speed decreases rapidly.

When an unexpected event occurs, the road speed exhibits a pattern in which the speed drops rapidly and then recovers. The proposed scheme applies a reduced weight and recovery weight to the speed decrease section and recovery section, respectively, to reduce the error in the primary predicted speed. First, the proposed scheme defines the decrease and recovery sections through the analysis of the relationship between the speed change and error rate. The proposed scheme uses the actual speed change and predicted speed change to define the criteria for identifying the decrease section. Furthermore, the proposed scheme uses the predicted speed change and actual historical speed change to apply event weights. The predicted speed change, historical actual speed change, and actual speed change are calculated using Equations (6)–(8), respectively. The term $\Delta PS_{T}^{t}$, which is the primary predicted speed change on the target road, refers to the difference between the predicted speed 15 min ago and that at the current time t. In this example, t is defined in 5 min intervals. When predicting the target road speed $\Delta HS_{T}^{t}$ at time t, the actual speeds that can be obtained are the data from 30 min in the past. Therefore, to predict $\Delta HS_{T}^{t}$, the proposed scheme calculates the difference in the actual speed between time t and a time 45 min in the past, as shown in Equation (7). Finally, the actual speed change $\Delta S_{T}^{t}$ indicates the difference between the actual speeds at time t and 15 min in the past.
(6)$$\Delta PS_{T}^{t}=PS_{T}^{t-3}-PS_{T}^{t}$$
(7)$$\Delta HS_{T}^{t}=S_{T}^{t-9}-S_{T}^{t-6}$$
(8)$$\Delta S_{T}^{t}=S_{T}^{t-3}-S_{T}^{t}$$

The proposed scheme defines the reference value of the speed decrease section using the error rate statistics of the predicted speed according to the speed change. The error rate of the predicted speed is calculated using Equation (9). When the error rate is close to one, the accuracy is high. The term $e_{T}^{t}$ denotes the error rate of the predicted speed on the target road at time t. A value greater than one indicates that the predicted speed is lower than the actual speed and a value less than one indicates that the predicted speed is higher than the actual speed.
(9)$$Error\;Rate\;{\left(e\right)}_{T}^{t}=S_{T}^{t}/PS_{T}^{t}\;\{\begin{array}{c} e>1\;:\;S_{T}^{t}>PS_{T}^{t}\\ e<1\;:\;S_{T}^{t}<PS_{T}^{t} \end{array}$$

The proposed scheme analyzes the effects of road speed changes on the prediction error rate based on events that occurred in the past to identify the speed decrease and recovery sections. The proposed scheme analyzes the speed decline trend based on changes in the primary predicted speed. However, the predicted speed follows the flow of the road speed in the future, as discussed in Section 3.4. Because it exhibits a gentler trend compared to the actual speed, we cannot determine the declining trend accurately. Therefore, the proposed scheme uses historical actual speed changes to identify rapid speed decrease sections.

Figure 9 shows error rate statistics according to the predicted speed changes analyzed using historical data. The orange bar graph indicates the predicted change in speed. When the predicted speed change is less than or equal to five, high accuracy is achieved with an error rate close to one. In contrast, when the predicted speed change increases to six or more, the error rate decreases. Furthermore, when the predicted speed change increases to 14 or more, the error rate increases to a value greater than one. This is because the road speed recovers immediately after a sharp decrease. Therefore, the proposed scheme uses a predicted speed change of six or more to define the first criterion for identifying a speed decrease section based on statistical analysis. However, because the predicted speed exhibits a gentler flow than the actual speed, the declining trend cannot be determined accurately. Therefore, the proposed scheme considers the actual speed changes to define the criteria for identifying speed decrease sections. The blue bar graph reshows the actual speed changes. The accuracy decreases when the actual speed change is between 4 and 10 and then increases and decreases repeatedly. Therefore, the proposed scheme defines the criterion for the second speed decrease section as an actual speed change of 10 or more.

The proposed scheme defines two criteria for identifying speed decrease sections based on error rate analysis according to the predicted speed and actual speed changes. If these criteria are met, the proposed scheme corrects the speed by considering the weight of the primary predicted speed. The weight is defined as 0.8, which is the average error rate of the two changes that represent the criteria for a rapid speed decrease section. Therefore, a 20% reduced speed is reflected in the predicted speed. The proposed scheme increases this weight after 30 min. The orange bar graph in Figure 9 demonstrates that the error rate increases in the recovery section. However, the blue graph of actual speed changes demonstrates that the prediction error rate decreases as the speed change increases. This indicates that the greater the speed decrease, the higher the predicted speed, which increases the difference. Therefore, the proposed scheme defines the decrease in weight (dω) differently according to the predicted speed change, as shown in Table 3.

Algorithm 1 shows the algorithm for applying weights to decrease the speed of sections. If the two criteria for identifying a speed decrease section are satisfied, then the proposed scheme changes the weight decrease, as shown in Table 3. Next, the proposed scheme modifies the primary predicted speed by reducing the weight and increases the count. A count of one reshows five minutes. After 30 min, the reduction in weight increases because the trend of the speed flow can be reflected accurately. Because a weight decrease of one is a meaningless value, the proposed scheme cannot reflect the decrease in weight when the weight decreases to one. Therefore, the proposed scheme stops reflecting the decrease in weight and executes the recovery weight application algorithm when the criteria for identifying a recovery section are satisfied, even if the criteria for identifying a decrease section are still satisfied.
**Algorithm 1:** Event Weighting Algorithm (Decrease Section)**Notation:**Speed Decrease Criteria1 = 6; Speed Decrease Criteria2 = 10;Decrease Weight (dw) = 0.8; Count = 1;**Input:** PSTt, ΔPSTt, ΔHSTt**Output:** PSTt′**if**ΔPSTt>Speed Decrease Criteria1 and ΔHSTt>Speed Decrease Criteria2
**then**
**check_recovery_criterial;****if**
ΔPSTt>ΔPSTt−1
**then**switch(int(ΔPSTt)) {case 7 :dω=0.7case 8 :dω=0.6case 9 :dω=0.5case 10 :dω=0.4
Count = 1;**end if**
PSTt′=PSTt ∗ dω;Count++; **end if**
**if** Count = 6 **then**dω=dω+0.1;Count = 1;**end if****if**
dω=1 **then****break;****end if****return**
PSTt′


The proposed scheme analyzes the relationship between the predicted speed change and actual speed to identify a speed recovery section. Figure 10 shows the relationship between the predicted speed change and actual speed. The red bar graph reveals that despite the fact that the primary predicted speed change increases sharply by more than 10, the predicted speed decreases in a gentle curve. The blue bar graph reveals that before the predicted speed change increases to 11, the actual speed has already transitioned from the negative direction to the positive direction and formed a recovery section. This means that the actual speed change, $\Delta HS_{T}^{t}$, changes from a positive number to a negative number. Therefore, when an event occurs, the road speed exhibits a sharp decrease and then recovers. In a section where the speed decreases, the historical actual speed change is a positive number, whereas in a section where the speed increases, the historical actual speed change is negative. Therefore, the proposed scheme identifies a recovery section by using the time point at which the historical actual speed change, $\Delta HS_{T}^{t}$, changes from a positive to a negative number.

When a speed recovery section is identified, the proposed scheme corrects the speed by applying the recovery weight. In contrast to a decrease in weight, the recovery weight is defined as 1.2 to correct the predicted speed with a 20% increase. Algorithm 2 shows the algorithm for applying the recovery section weight. If the historical speed change becomes a negative number, then the proposed scheme identifies a recovery section. If the recovery weight is greater than one, it applies the recovery weight to the primary predicted speed to correct the speed. In the recovery section, the speed is recovered with a relatively gentle curve. Because the primary predicted speed can reflect the trend of the recovery section after 30 min, the proposed scheme reduces the weight by 0.1 in each iteration after 30 min.
**Algorithm 2:** Event Weighting Algorithm (Recovery Section)**Notation:**Recovery Weight (rw) = 0.2;Count = 0;**Input:**PSTt**Output:**PSTt′ **if**  Count = 6 **then**rω=rω−0.1;Count = 0;

**end if**

$\mathrm{if}\;HS_{T}^{t}<0\;\mathrm{and}\;r\omega >1.0\;\mathrm{then}$
$PS_{T}^{t}{}^{\prime }=PS_{T}^{t}*r\omega ;$
Count++;

**end if**

**return**
$PS_{T}^{t}{}^{\prime }$


## 4. Performance Evaluation

### 4.1. Performance Evaluation Environment

In this paper, we demonstrate the excellent performance of the proposed road speed prediction scheme by comparing its performance to that of existing schemes. Table 4 shows performance evaluation environments. We conducted performance evaluations on a PC with an Intel Core i5-4440 at 3.10 GHz CPU, 8.00 GB of RAM, and Windows 10 operating system. The proposed scheme was implemented using the Python language and Keras library [26] in the Python Anaconda custom environment. Table 5 shows datasets that are used for performance evaluation. For the speed data used in our performance evaluations, we collected sectional travel speed data from the collection system of the vehicle detection system provided on the open expressway data portal site by the Korea Expressway Corporation. For weather data, we collected rainfall data from the disaster prevention and weather observation data provided on the Open MET Data Portal of the Korea Metrological Administration. For performance evaluations, we used speed data and rainfall data collected from 24 June to 1 September 2020 as the training dataset and used speed data and rainfall data collected from 2 September to 6 October 2020 as the prediction dataset.

The performance evaluation of the proposed road speed prediction scheme mainly consisted of a standalone performance evaluation and a comparative performance evaluation considering existing schemes. The standalone performance evaluation verified the validity of each type of road environment data considered in the proposed scheme by comparing the results of applying and not applying the following data types: rainfall data, historical average speed data used to reflect the normal flow of the road, and event weights used to predict sharp or unusual changes in road flow.

We selected two schemes for comparative performance evaluations. The first scheme is that proposed in [15], which predicts the near-future speed on the target road by considering the effects of the speeds of neighboring roads on the speed of the target road. We denote the scheme from [15] as the TN-P scheme. The second scheme is that proposed in [21], which predicts road congestion using an RNN. We denote the scheme from [21] as the RNN-P scheme. To evaluate prediction accuracy in this study, we calculated the RMSEs between the predicted and actual speeds. As the RMSE decreases, prediction accuracy increases.

### 4.2. Standalone Performance Evaluation

#### 4.2.1. Results Obtained by Reflecting Weather

The proposed scheme uses weather data to reflect weather effects. The proposed scheme trains a prediction model with optimal weights to reflect the effects of rainfall and the speed of the neighboring roads on the speed of the target road. A prediction model is used to estimate the primary speed. In this study, we conducted performance evaluations by comparing the results between cases when rainfall data were reflected (Prediction Speed A) and when they were not reflected (Prediction Speed B). The case that did not reflect rainfall data corresponds to the scheme proposed in [15]. In our performance evaluations, we used the predicted speed results for a rainy day from 12:00 to 8:00 p.m. on 5 September 2020.

Figure 11 shows the speed predicted considering the effects of rainfall, actual speed, and RMSE. Because the prediction model considering rainfall data was used, the predicted speed is similar to the actual speed as a result of considering the effects of rainfall using expected rainfall data. Figure 12 shows the speed predicted considering the effects of neighboring roads only, the actual speed, and RMSE. In a section where the speed decreases as a result of effects of rainfall, this effect is reflected according to a decreasing trend in the speed, but because the effects of rainfall are not reflected, the RMSE is higher than that in Figure 11. In Figure 11 and Figure 12, we provide the results on a specific date. However, we confirmed that similar results are shown when performing experiments on different dates.

Figure 13 shows the (A) RMSE value considering the effects of rainfall and the (B) RMSE value that does not consider the effects of rainfall. We calculated the RMSEs using data from 12:00 to 8:00 p.m. on 5 September 2020. When the RMSE is close to zero, the accuracy is high. The value in A is 3.66, which is 0.29 less than that in B, in which the value is 3.95. Therefore, it is confirmed that higher accuracy is achieved when rainfall data are reflected. In addition, we confirmed that the experimental results using rainfall data for different dates showed higher accuracy than the experimental results without using rainfall data.

#### 4.2.2. Results Obtained by Reflecting Historical Average Speeds

Long-term road speed prediction using an LSTM model does not predict sections in which the flow on a road decreases on average because the data from the previous time point just before the prediction time point have the largest influence. Therefore, the proposed scheme defines the average speed section by performing statistical analysis on historical average speed changes to reflect the normal flow of the target road. Furthermore, the proposed scheme reflects historical average speeds in road speed predictions through comparisons to the primary predicted speed. In this study, we conducted standalone performance evaluations by comparing the results between when the historical average speed was reflected (Prediction Speed A) and when it was not reflected (Prediction Speed B). For the case of not reflecting the historical average speed, we used the proposed scheme up to the primary speed prediction stage.

Figure 14 shows the actual speed, prediction result considering the historical average speed (Prediction Speed A), and RMSE from 5:30 to 7:30 a.m. on 9 September 2020. Because Prediction Speed A reflects the historical average speed between 5:40 and 6:25 a.m., the RMSE value decreases significantly, meaning the predicted speed moves closer to the actual speed. Figure 15 shows the actual speed, prediction result without considering the historical average speed (Prediction Speed B), and RMSE. Figure 15 exhibits a relatively higher RMSE than A because B does not consider the average flow of the morning rush hour and follows the decreasing trend of the speed at a later time. In Figure 14 and Figure 15, we provide the results on a specific date. However, we confirmed that the similar results are shown when performing experiments on different dates.

Figure 16 shows the (A) average RMSE considering the historical average speed and the (B) average RMSE without considering the historical average speed. The RMSE was calculated using the data from the section in which the historical average speed was reflected between 5:40 and 6:25 a.m. on 9 September 2020. Prediction Speed A is 7.76 with an RMSE value reduced by 4.34 compared to Prediction Speed B, which is 12.10. Therefore, it is confirmed that higher accuracy is obtained when the historical average speed is considered. In addition, we confirmed that the experimental results using historical average speed data for different dates showed higher accuracy than the experimental results without using historical average speed data.

#### 4.2.3. Results Obtained by Reflecting Event Weights

The proposed scheme applies event weights to sections in which the speed changes rapidly to reflect an uncommon flow on the road. The proposed scheme defines the criteria for identifying a decrease–recovery pattern, which occurs when the speed changes rapidly, by statistically analyzing speed changes. Based on this analysis, the decrease in weight and recovery weight are defined. Therefore, the proposed scheme improves prediction accuracy in sections where the speed changes rapidly in response to an event such as a traffic accident or construction work. In this study, we conducted standalone performance evaluations by comparing the results between when the event weight was applied and when it was not applied. For the case of not applying the event weight, we used the proposed scheme up to the primary speed prediction stage. For this performance evaluation, we used the speed prediction results from 7:00 a.m. to 7:00 p.m. on 9 September 2020.

Figure 17 shows the predicted speed considering the event weight (Prediction Speed A), actual speed, and RMSE. Because the decrease in weight is applied by identifying the speed decrease section, the accuracy of Prediction Speed A is high in the road speed section where the speed decreases rapidly. Furthermore, Prediction Speed A also exhibits high accuracy in the speed recovery section because the recovery weight is applied. Figure 18 shows the predicted speed without considering the event weight (prediction speed B), actual speed, and RMSE. Prediction Speed B exhibits a trend of following the sections in which the road speed decreases rapidly and recovers because the speed data from 30 min ago have the greatest influence. Additionally, Prediction Speed B exhibits low accuracy because the speed in the rapidly decreasing section is predicted with a gentle flow, which does not match the actual speed flow. In Figure 17 and Figure 18, we provide the results on a specific date. However, we confirmed that similar results are found when performing experiments on different dates.

Figure 19 shows the (A) average RMSE value considering the event weight and (B) average RMSE value without considering the event weight. For this performance evaluation, we classified two sections in which the road speed decreased sharply and calculated the average RMSE values. Section 1 refers to the section in which the speed change occurred in the morning. Section 2 refers to the section in which the speed change occurred in the afternoon. The RMSE in A in the morning section is 11.80 with an RMSE reduction of 60.31 compared to B, which is 72.11. Overall, the prediction error decreases significantly when applying the event weight. The RMSE in A in the afternoon section is 14.67 with an RMSE reduction of 8.33 compared to B, in which the RMSE value is 23.00. These performance evaluation results confirm that a higher accuracy is obtained when the event weight is considered. In addition, we confirmed that the experimental results using the event weight for different dates showed higher accuracy than the experimental results without using event weight.

### 4.3. Performance Comparison

#### 4.3.1. Performance Comparison between the TN-P Scheme and Proposed Scheme

The TN-P scheme predicts the near-future speed on the target road with higher accuracy compared to the case of considering only the speed of the target road because it considers the effects of the speeds of neighboring roads on the speed of the target road. For performance comparisons, we used the TN-P scheme to predict the road speed 30 min in the future on 9 September 2019.

Figure 20 shows the results of predicting the road speed after 30 min using the TN-P scheme, actual speed, and RMSE. In the TN-P scheme, because the road speed data from 30 min ago have the largest influence, the flow is predicted with a delay, increasing the overall error. In particular, the RMSE value increases significantly in the section where the road speed changes rapidly. Figure 21 shows the results of using the proposed scheme to predict the road speed after 30 min, actual speed, and RMSE. The proposed scheme considers all of the following factors: the effects of neighboring roads, effects of weather, typical flow of the target road based on historical average speeds, and event weights based on rapidly changing flows on the road. Because it did not rain on 9 September, weather had no effect. When the performance evaluation results were compared between the proposed scheme and TN-P scheme, we found that the accuracy was higher for the proposed scheme that considers the typical flow in which the road speed decreases rapidly during the morning rush hour. Furthermore, the proposed scheme exhibits higher accuracy than the TN-P scheme because it applies event weights to sections in which the road speed decreases rapidly between 8:00 and 11:00 a.m., and between 3:00 and 6:00 p.m., after which the speed recovers. The RMSE value in Figure 21 is lower than that in Figure 20 overall. In particular, the RMSE is significantly lower in Figure 20 in the sections where the speed changes rapidly. When we calculated the average RMSE values of the section in which the proposed scheme was applied for 9 September 2019, the average RMSE was 30.81 with a reduction of 76.4 compared to the average RMSE of the TN-P scheme, which was 107.21. These performance evaluation results confirm that the proposed scheme performs better than the TN-P scheme. In Figure 20 and Figure 21, we provide the results on a specific date. However, we confirmed that the similar results are shown when performing experiments on different dates.

#### 4.3.2. Performance Comparison between the RNN-P Scheme and Proposed Scheme

We compared the performance of the proposed scheme to that of road congestion prediction using an RNN [21] to evaluate the performance of the LSTM as a learning tool for primary speed prediction in the proposed scheme. The RNN is a machine learning scheme that is widely used for time series predictions because it considers the connectivity of time. However, RNN then faces the vanishing gradient problem, where the effects of initial data disappear as the time gap increases. Therefore, the proposed scheme adopts LSTM, which is designed to overcome the vanishing gradient problem, as a learning tool for primary speed prediction.

Figure 22 shows the RMSE (RNN-P) value of the speed predicted using the RNN and the RMSE (LSTM-P) value of the speed predicted using the LSTM. We calculated the average RMSE of the predicted speeds on a daily basis from 8 September to 14 September 2020. In the results for all days excluding Monday and Tuesday, the RMSE of the speed predicted using the LSTM-P is low, resulting in high average accuracy. This confirms the superiority of the proposed scheme. In addition, we confirmed that the experimental results using LSTM for different dates showed higher accuracy than the experimental results using RNN.

## 5. Conclusions

This paper proposed a road speed prediction scheme based on the analysis of various factors affecting road speed. The proposed scheme uses rainfall data from the target area and speed data collected from connected roads to reflect the effects of weather and connected roads. The proposed scheme trains an LSTM prediction model with optimal weights through an iterative training process. The proposed scheme uses a prediction model to predict primary speeds by reflecting the effects of rainfall and connected roads. The proposed scheme then applies historical average speed data by analyzing average speed changes to reflect the typical flow of the target road. Finally, the proposed scheme analyzes speed changes to predict traffic flows that change rapidly when an event occurs. Based on the analyzed results, criteria for identifying speed decrease and recovery sections are defined, and an event weight was defined. Finally, the effects of events are reflected in the predictions of road speeds. The proposed scheme improves accuracy by approximately 7% compared to existing schemes that consider only the effects of connected roads. In areas with rain, although the accuracy increase rate is relatively small because the impact is calculated not only during a specific rain period, but also over a wide range of time, a comparison between the graphs in Figure 11, Figure 12 and Figure 13 revealed a significant decrease in the error rate. The performance evaluation results on non-rainy days revealed that the accuracy is approximately 83% higher for the proposed scheme compared to existing schemes. The proposed scheme exhibits high accuracy because it reflects the rapidly changing flows of the target road by analyzing data that affect road speeds.

In our future works, we will conduct performance evaluations by applying various road speed prediction schemes that use speed change as input and directly predict the speed change. We will also conduct additional performance evaluations between deep-learning-based models and statistical baseline models to show the superiority of the proposed scheme. Furthermore, we will investigate additional factors that affect road speed to apply them to services that provide real-time road situation information. Finally, we will develop case studies using the proposed scheme.

## Figures and Tables

**Figure 1 sensors-22-02606-f001:**
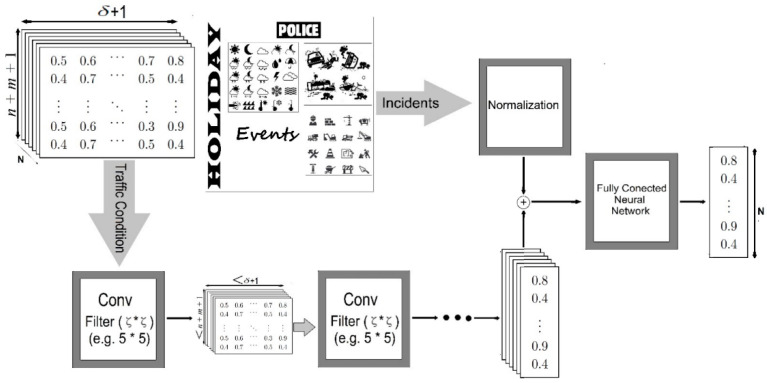
Overall processing structure of the road congestion prediction model.

**Figure 2 sensors-22-02606-f002:**
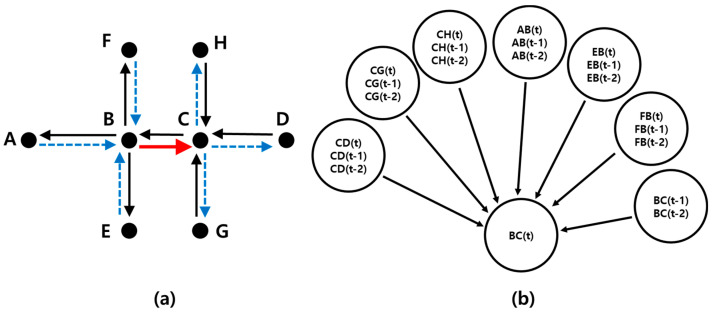
Road network and Bayesian network designed to predict road speeds. (**a**) Example of a road network. (**b**) Bayesian network designed to predict road BC.

**Figure 3 sensors-22-02606-f003:**
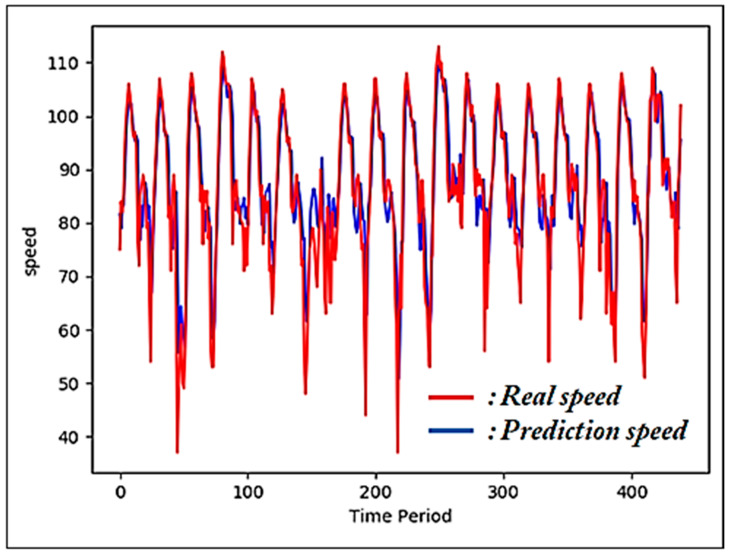
Graph comparing the actual speeds and predicted speeds considering the effects of the speeds of connected roads.

**Figure 4 sensors-22-02606-f004:**
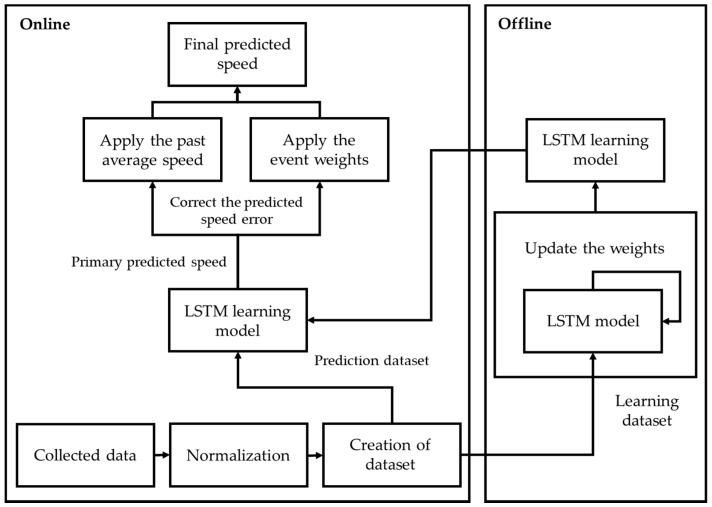
Overall structure of the proposed road speed prediction scheme.

**Figure 5 sensors-22-02606-f005:**
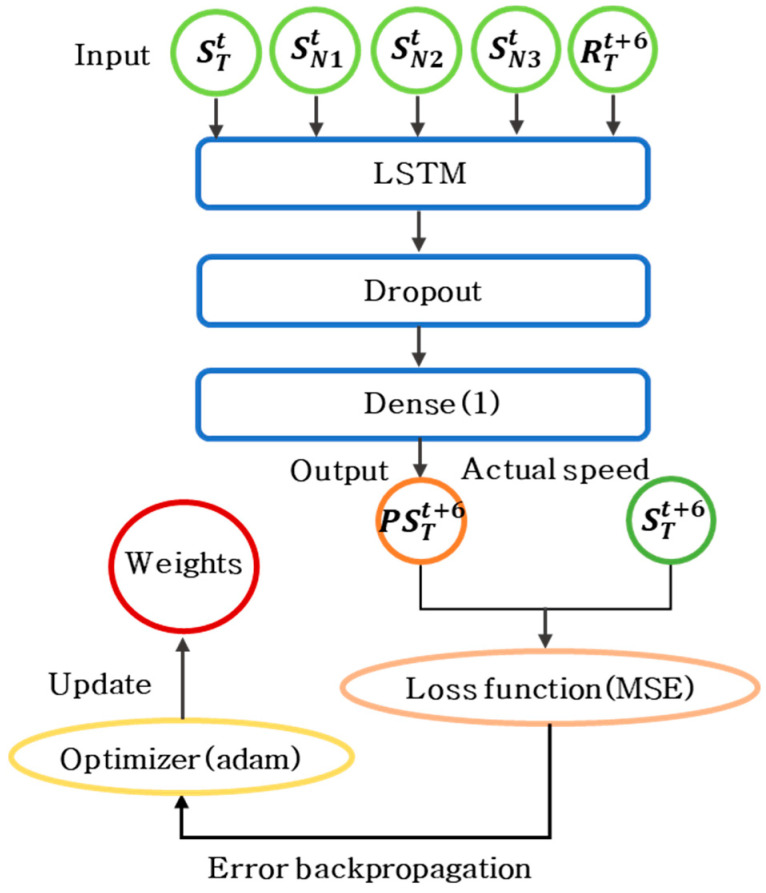
Training process of LSTM model.

**Figure 6 sensors-22-02606-f006:**
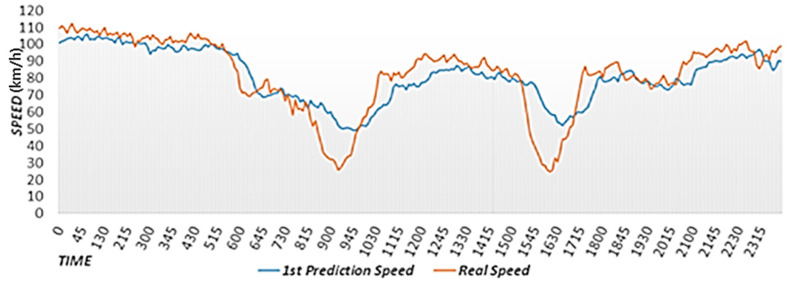
Graph of actual road speeds and primary predicted speeds over time.

**Figure 7 sensors-22-02606-f007:**
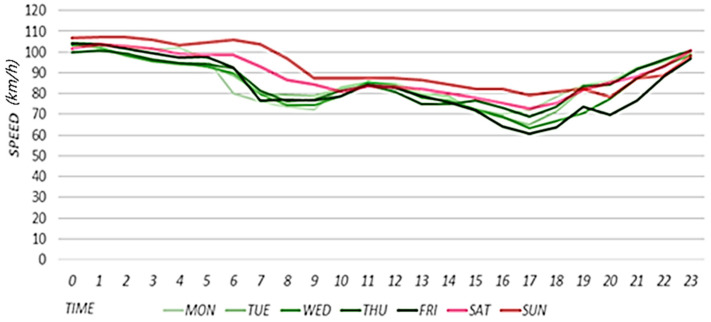
Results of calculating the average speed of the target road.

**Figure 8 sensors-22-02606-f008:**
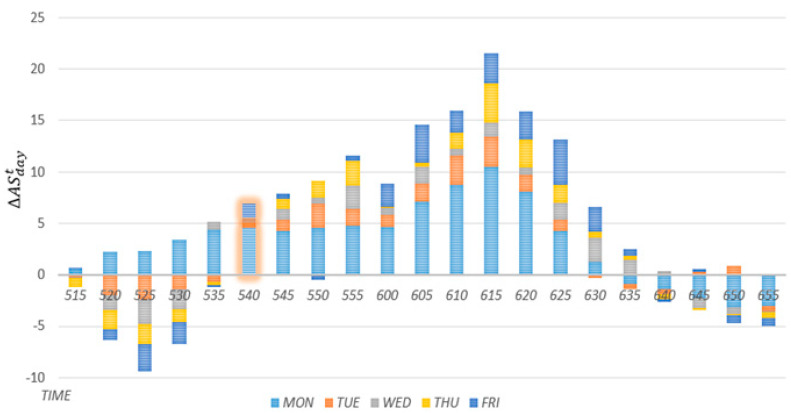
Historical speed changes during the rush hours for each day of the week.

**Figure 9 sensors-22-02606-f009:**
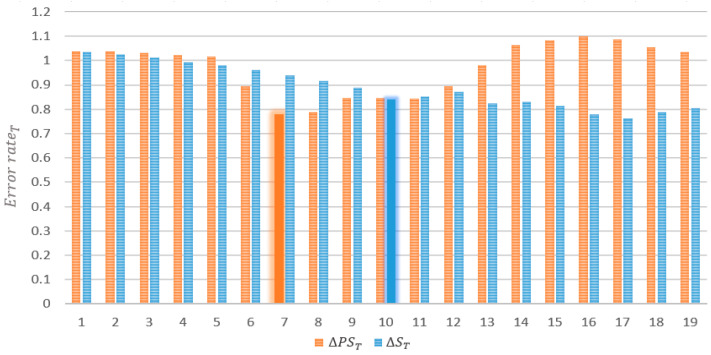
Error rate statistics according to the predicted speed changes.

**Figure 10 sensors-22-02606-f010:**
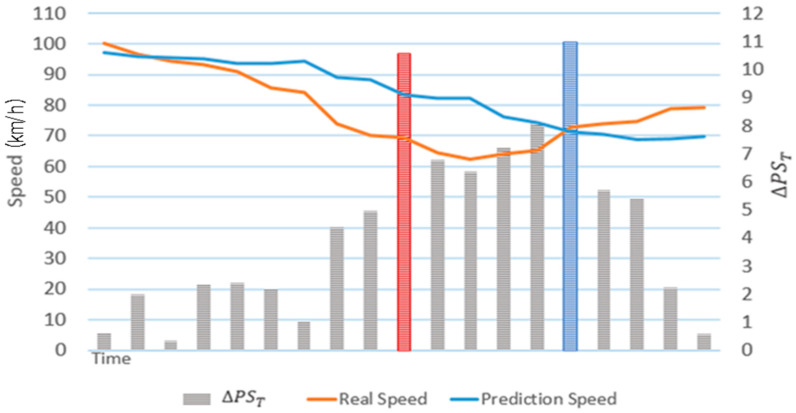
Relationship between the predicted speed change and actual speed.

**Figure 11 sensors-22-02606-f011:**
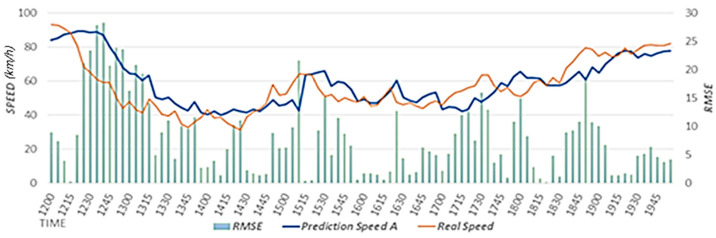
Speed predicted considering the effects of rainfall.

**Figure 12 sensors-22-02606-f012:**
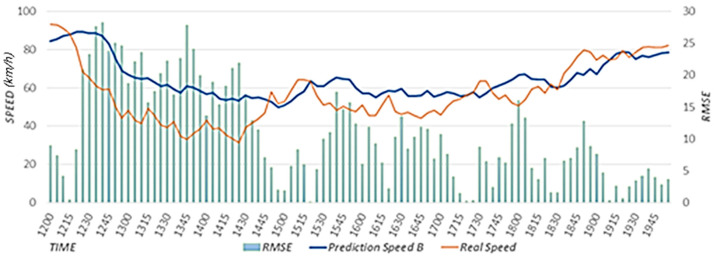
Speed predicted considering the effects of neighboring roads only.

**Figure 13 sensors-22-02606-f013:**
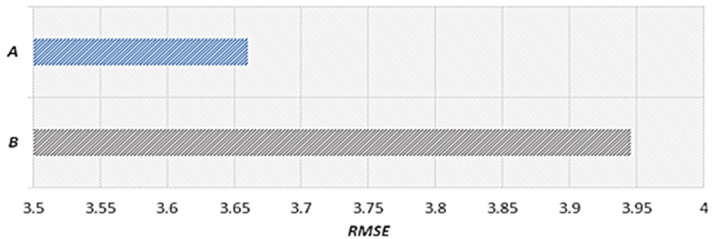
RMSE value considering the effects of rainfall. (**A**) RMSE value considering the effects of rainfall. (**B**) RMSE value that does not consider the effects of rainfall.

**Figure 14 sensors-22-02606-f014:**
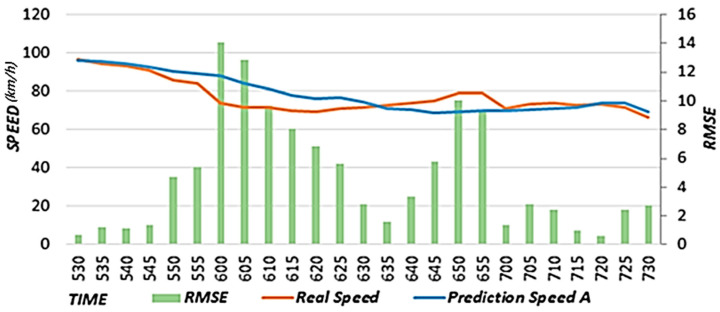
Prediction result considering the historical average speed.

**Figure 15 sensors-22-02606-f015:**
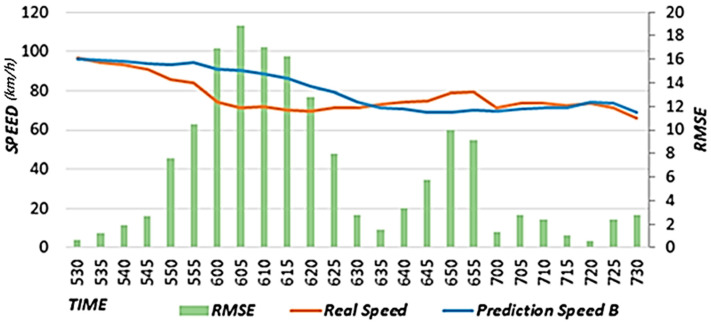
Prediction result without considering the historical average speed.

**Figure 16 sensors-22-02606-f016:**
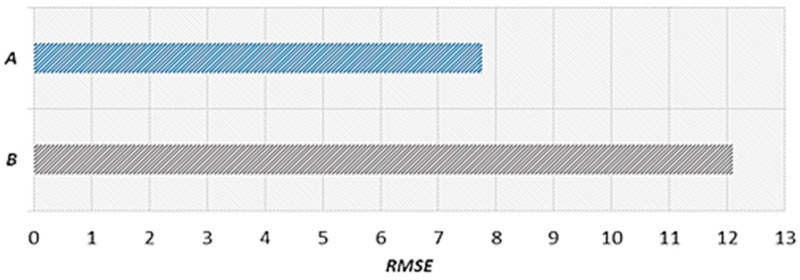
RMSE value considering the historical average speed. (**A**) average RMSE considering the historical average speed. (**B**) average RMSE without considering the historical average speed.

**Figure 17 sensors-22-02606-f017:**
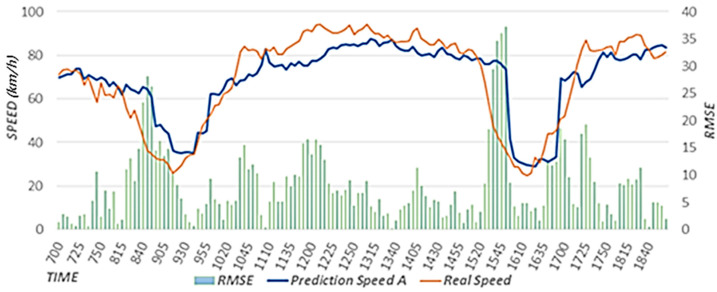
Predicted speed considering the event weight.

**Figure 18 sensors-22-02606-f018:**
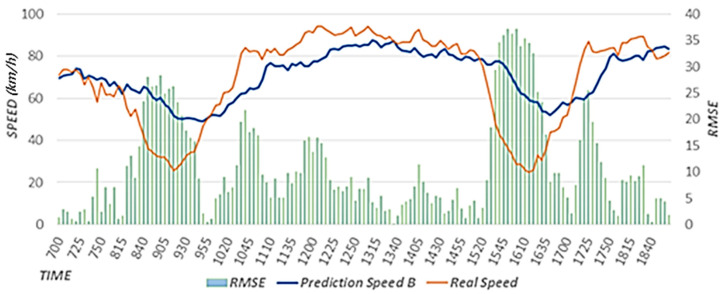
Predicted speed without considering the event weight.

**Figure 19 sensors-22-02606-f019:**
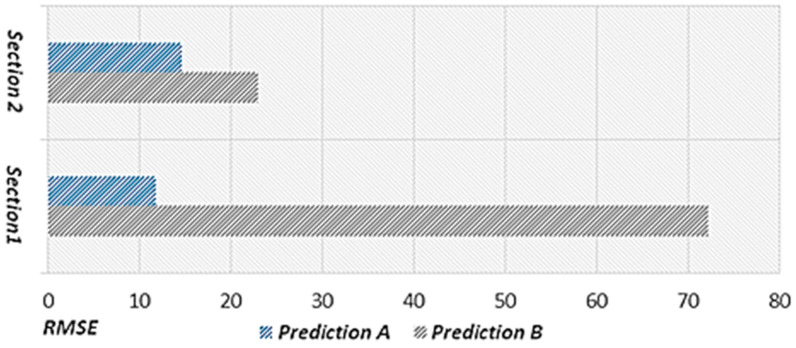
RMSE value considering the event weight. (Section 1) the section in which the speed change occurred in the morning. (Section 2) the section in which the speed change occurred in the afternoon.

**Figure 20 sensors-22-02606-f020:**
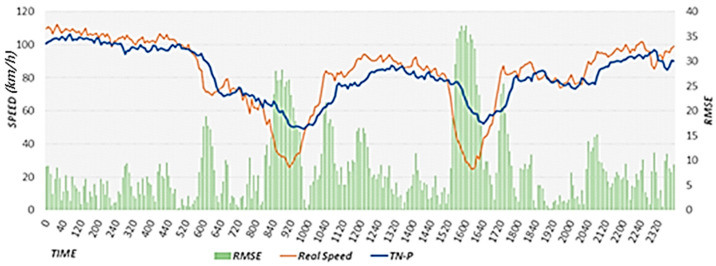
Results of predicting the road speed using the TN-P scheme.

**Figure 21 sensors-22-02606-f021:**
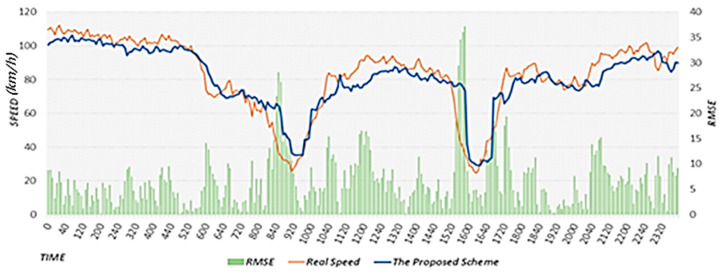
Results of predicting the road speed using the proposed scheme.

**Figure 22 sensors-22-02606-f022:**
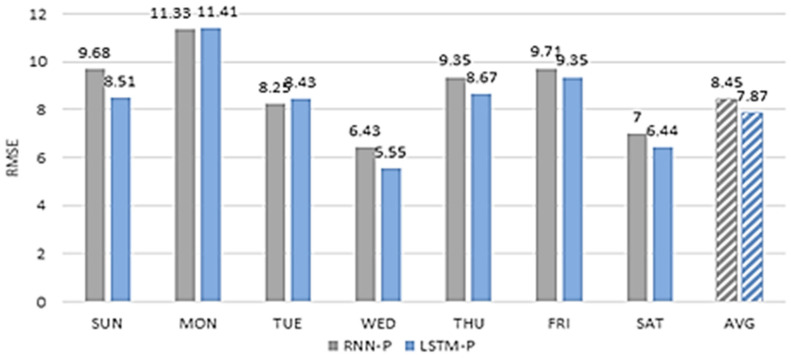
RMSE value of the speed predicted using the RNN and the LSTM.

**Table 1 sensors-22-02606-t001:** Example prediction dataset.

	$PRF_{T}^{i+6}$	$S_{N_{1}}^{i}$	$S_{N_{2}}^{i}$	$S_{N_{3}}^{i}$	$S_{T}^{i}$
t	0.214334	0.870698	0.905612	0.848839	0.851250
t + 1	0.239343	0.866772	0.907388	0.855186	0.845409
t + 2	0.0	0.862847	0.909164	0.861533	0.839569
t + 3	0.0	0.906615	0.917066	0.910139	0.877798
t + 4	0.0	0.880805	0.910407	0.865375	0.829195
t + 5	0.0	0.851535	0.942584	0.845357	0.842598
t + 6	0.0	0.902145	0.920457	0.864527	0.864215
t + 7	0.0	0.894255	0.901565	0.902145	0.854112
t + 8	0.0	0.904232	0.920545	0.901565	0.945515

**Table 2 sensors-22-02606-t002:** Example training dataset.

	$PRF_{T}^{i+6}$	$S_{N_{1}}^{i}$	$S_{N_{2}}^{i}$	$S_{N_{3}}^{i}$	$S_{T}^{i}$	$S_{T}^{i+6}$
t	0.214334	0.870698	0.905612	0.848839	0.851250	0.864215
t + 1	0.239343	0.866772	0.907388	0.855186	0.845409	0.854112
t + 2	0.0	0.862847	0.909164	0.861533	0.839569	0.945515
t + 3	0.0	0.906615	0.917066	0.910139	0.877798	0.934532
t + 4	0.0	0.880805	0.910407	0.865375	0.829195	0.928745
t + 5	0.0	0.851535	0.942584	0.845357	0.842598	0.843523
t + 6	0.0	0.902145	0.920457	0.864527	0.864215	0.874353
t + 7	0.0	0.894255	0.901565	0.902145	0.854112	0.892345
t + 8	0.0	0.904232	0.920545	0.901565	0.945515	0.902343

**Table 3 sensors-22-02606-t003:** Decrease in weight differently defined according to the predicted speed change.

$\Delta PS_{T}^{t}$	dω
6	0.8
7	0.7
8	0.6
9	0.5
10	0.4
11	0.3
12	0.2
13	0.1

**Table 4 sensors-22-02606-t004:** Evaluation environments.

Category	Description
Processor	Intel(R) Core(TM) i5-4440K 3.10 GHz 4 Core
Memory	8.0 GB
Operating system	Windows 10
Language used	Python 3
Platform used	Python 3.5.6 Anaconda custom

**Table 5 sensors-22-02606-t005:** Datasets.

Category	Collection Period	Size
Training dataset	24 June 202–1 September 2020	20,160 cases
Prediction dataset	2 September 2020–6 October 2020	10,080 cases

## Data Availability

Not applicable.

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
