# Peer review of "Road Speed Prediction Scheme by Analyzing Road Environment Data"

_sensors, 2022, doi:10.3390/s22072606_

Round 1

Reviewer 1 Report

  1. It is not clear the intrinsic relationship between traffic speed prediction and traffic congestion considering the statement “Therefore, we can reduce the occurrence of traffic congestion by predicting the road speed.” Please add more explanations.
  2. It was not clear did the author predict microscopic or macroscopic speed? Did the authors predict individual or average speed?
  3. Please illustrate the unique advantages of using LSTM model to predict speeds due to that many prediction-relevant models are available.
  4. Please illustrate figure 2 in more detail.
  5. The authors were recommended to additional information about data samples used in the study (e.g., for figure 3).
  6. The following studies were recommended to be properly cited: [1] Traffic flow prediction by an ensemble framework with data denoising and deep learning model, Physica A: Statistical Mechanics and its Applications, vol. 565, p. 125574, 2021. [2]Personalized Vehicle Trajectory Prediction Based on Joint Time-Series Modeling for Connected Vehicles, IEEE Transactions on Vehicular Technology, vol. 69, no. 2, pp. 1341-1352.

Reviewer 2 Report

This is an interesting topic and of course subject of research by many academics and practitioners. The following comments are meant to improve the quality and readability of your work:

Make sure grammatical mistakes and typos are taken care of.

Describe your case study and provide a descriptive analysis of your speed dataset.

Make sure claims about use of events in prediction model are well supported in the text.

It is difficult to draw strong conclusions based on a limited number of examples. To ensure consistent performance of the proposed method more tests and case studies are needed.  

Round 2

Reviewer 1 Report

My comments have been addressed. 

Author Response

Dear Reviewer,

Many thanks for your good comments.

Best regards,

Jaesoo Yoo

This manuscript is a resubmission of an earlier submission. The following is a list of the peer review reports and author responses from that submission.

Round 1

Reviewer 1 Report

This paper proposed an LSTM-based framework for speed prediction into the future of 30 minutes based on a 5-min interval. This framework takes as input the historical speed data of the target road and the neighboring roads, as well as the rainfall data for predicting the primary speed of the target road. Then, the primary prediction is post-processed by considering historical average speed, speed change of the historical speed, as well as the predicted speed, and unexpected events to calibrate the primary prediction and improve the prediction accuracy. Compared to an RNN-based baseline, the LSTM-based framework achieved enhanced performance.  One major drawback of the approach is that the pose-processing/correction is very data-depend, i.e., the authors used statistics from the Korea Expressway data. In other words, these statistics may vary from other datasets, and it is not clear whether this post-processing can be easily generalized. Also, this pose-processing is designed in a very specific way to correct the LSTM-based model failing to predict rapid changes of the speeds, rather than addressing this problem by exploring the network architecture and the input features. Given the very limited generalizability potential, I would, unfortunately, reject this work for a publication. On the other hand, there is a big potential to improve this work. Here are the detailed comments.

This paper has shown that speed change is a very important factor for speed prediction. I would recommend the authors to consider using speed change as input and directly predict the speed change, instead of predicting the actual speed. The advantage of predicting such type of “offset” has been already proven in many other domains.

There might be a solution to incorporate historical average speed as a feature for the prediction task as an end-to-end training, instead of post-processing. Also, given the very similar performance between RNN-P and LSTM-P measured by RMSE (Fig. 24), there should be a comparison between the DL-based models and a statistical baseline model just using the historical data. This can help validate whether there is a clear gain from the DL methods.

There are several aspects of the LSTM-based model needed to be clarified. First, the sequence length of the LSTM is not clear, i.e., are the feature sequences given in a time-series format of {t_i, t_i+1, …, t_i+6}, or just {t_i, t_i+6}? Purely based on Figure 5, it seems that the sequential speed data is not fully manipulated (if I am wrong, please correct me). How are the sequences partitioned over the 24 hours a day, is there a sliding window method or any overlap between two consecutive sequences? In addition, why is there no analysis of the error between PRF and rainfall (e.g., Table 1 and Table 2), are they also included into the loss function? Moreover, it might be helpful to explain why the time interval and the prediction horizon are set to 5 and 30 minutes, respectively.

The resolution of many figures in this paper needs to be increased for a better visualization. In Fig. 5, it might be better to point out that those “weights” are trainable weights in the LSTM-based model, as “weights” are also used to refer to the manually selected weights for the correction schemes, e.g., Table3. Also, Fig. 15, 18, 21 could be plotted in one figure (or presented as a table) to show the error scales across the different correction schemes. There is a text overlap in Table 4, (which might be caused by the pdf compiler).

Please check if the citation [19] on P5 L199 is correct.     

Reviewer 2 Report

The topic of study is interesting, unfortunately, many unprofessional expressions lead to poor readability of the paper, and thus difficult to track the research method and results.

A professional editor is strongly recommended to proofread or rewrite the paper. Problems, for example, are repetitive (e.g., line 26-27 vs line 31-32), inaccurate (e.g., line 16: the occurrence of traffic congestion can be reduced by predicting road speed), awkward (e.g., line 87: on very rainy days, most vehicles drive at reduced speeds, line 98: data arriving near the time of prediction have the greatest impact on road speed prediction), and a lot of unprofessional expressions, just name a few.

The authors may rethink the structure and layout of the paper. For example, lines 73-104 may be split into several paragraphs.

Almost all the figures are not of high quality. For example, Figure 1 is a blur and is a direct snapshot from literature, which is not recommended in a professional article.

Equations (1) and (2) are not expressed in a professional way.

What do the values in Tables 1 and 2 refer to? Any description provided, any unit? Is the value in six-digit accuracy really necessary?

What is the contribution of the study, especially compared to the reference (8)?

For the entire study of speed, what is the unit of the speed?